# Bacterial Ribosome Rescue Systems

**DOI:** 10.3390/microorganisms10020372

**Published:** 2022-02-05

**Authors:** Daisuke Kurita, Hyouta Himeno

**Affiliations:** Department of Biochemistry and Molecular Biology, Hirosaki University, 3, Bunkyo-cho, Hirosaki 036-8561, Japan; dkurita@hirosaki-u.ac.jp

**Keywords:** ribosome, ribosome rescue system, protein synthesis

## Abstract

To maintain proteostasis, the cell employs multiple ribosome rescue systems to relieve the stalled ribosome on problematic mRNA. One example of problematic mRNA is non-stop mRNA that lacks an in-frame stop codon produced by endonucleolytic cleavage or transcription error. In *Escherichia coli*, there are at least three ribosome rescue systems that deal with the ribosome stalled on non-stop mRNA. According to one estimation, 2–4% of translation is the target of ribosome rescue systems even under normal growth conditions. In the present review, we discuss the recent findings of ribosome rescue systems in bacteria.

## 1. Introduction

Ribosomes often stall during protein biosynthesis. *Trans*-translation was first identified as the system to rescue such stalled ribosomes [1,2]. This system almost completely distributed among the bacterial domain, whereas it is not essential for survival of many bacteria. It might be true that *trans*-translation is the main system, but we came to know that the bacterial ribosome rescue systems are more diversified than we thought.

## 2. *Trans*-Translation

In the middle of 1990s, tmRNA (SsrA) was found to have a tRNA-like structure, which accepts L-alanine at the 3′ end like tRNA, and then it was found to contain a coding sequence between the first two of four pseudoknot structures in the middle of this RNA [3,4,5]. By elaborately collaborating the tRNA and mRNA functions, tmRNA together with its partner protein SmpB facilitates a novel translation to rescue the stalled ribosome [6]. Ala-tmRNA in complex with SmpB and EF-Tu enters the vacant A-site of the stalled ribosome on a 3′ truncated mRNA (nonstop mRNA) like an aminoacyl-tRNA and thereafter the coding region acts as an mRNA, by which translation is allowed to resume (Figure 1). In this process, the tRNA-like domain of tmRNA and the globular domain of SmpB mimics the upper and lower halves of tRNA, respectively, and the unstructured C-terminal tail of SmpB recognizes the vacant mRNA entry channel of the stalled ribosome [7,8,9,10]. As a consequence of this irregular translation, called *trans*-translation, a peptide tag that acts as a degradation signal by ATP-dependent proteases such as ClpXP is attached to the truncated C-termini of the nascent polypeptides, so that the *trans*-translation products do not accumulate in the cell [11]. tmRNA and SmpB are almost completely conserved among the bacterial domain, while they are not found in eukaryotes with the exception of some chloroplasts [12].

tmRNA with SmpB preferentially recognizes nonstop mRNA for *trans*-translation. To avoid repetition of *trans*-translation on a nonstop mRNA, *trans*-translation itself further promotes degradation of nonstop mRNA [13]. RNase R, a 3′ to 5′ exoribonuclease, degrades nonstop mRNA coupled with *trans*-translation [14,15,16,17,18]. A lysine residue in the C-terminal region of *E. coli* RNase R is acetylated in the exponential phase, resulting in the phase-specific degradation of RNase R [19].

Structural studies revealed that the universally conserved nucleotides forming the decoding center, G530, A1492 and A1493 of 16S rRNA and A1913 of 23S rRNA undergo conformational rearrangements in *trans*-translation as well as canonical translation [10,20,21]. In free 30S subunit, A1492 and A1493 are stacked in the interior of helix 44 (h44) of 16S rRNA, while G530 adopts a *syn* conformation (Figure 2) [22]. Upon IF1 binding, the stacking interaction between A1492 and A1493 is disrupted; A1492 is sandwiched between IF1 and ribosomal protein uS12, while A1493 is buried in a pocket on IF1 [23]. After docking of the 30S subunit to the 50S subunit, A1492 and A1493 again form stacking interaction on each other and adopt partially flipped-out conformation [24]. A1493 is further stabilized by stacking with A1913 of Helix 69 (H69) of 23S rRNA. In translation elongation, A1492 and A1493 become fully flipped out of h44 to monitor the geometry of the codon-anticodon helix and G530 rotates from a *syn*- to an *anti*-conformation [25].

In pre-accommodation of tmRNA·SmpB, A1492 and A1493 become unstacked and in a flipped-in state, while G530 adopts a *syn* conformation [21]. After accommodation, A1493 becomes flipped-out to stack with SmpB His22, while A1492 stacks on A1913 in the interior of h44. Then EF-G promotes translocation of tmRNA·SmpB from the A-site to the P-site. In this translocated state, the first nucleotide of the resume codon on tmRNA (G90 in *E. coli*) is stacked on A1493. This stacking interaction might be the reason why the purine base at the first nucleotide of the resume codon is conserved in all bacteria. This is consistent with a mutation study, in which G90C or G90U substitution decreases the efficiency of tagging activity in vitro, while G90A does not [26]. While A1492 remains stacking on A1913 within h44, G530 interacts with the ribose of the third nucleotide of the resume codon (A92 in *E. coli*).

## 3. RF-Dependent Ribosome Rescue Factors

As described above, every bacterial cell has the *trans*-translation system, whereas it is not essential in many bacteria [27,28,29,30]. This raised the possibility that the bacterial cell has an alternative ribosome rescue system. In 2010, a new kind of ribosome rescue factor named ArfA (alternative ribosome rescue factor A) was identified [31]. Deletion of this gene from the genome of *E. coli* strain lacking the tmRNA gene (∆ssrA) causes a synthetic lethal phenotype. Ribosome rescue by ArfA requires a peptide release factor, RF2 [32,33]. At the end of protein synthesis, RF2 as well as RF1 enters the A-site to recognize a stop codon and hydrolyzes the peptidyl-tRNA in the P-site to release the completed polypeptide from the translating ribosome. RF2 also enters the A-site free of mRNA only when it is with ArfA and hydrolyzes the peptidyl-tRNA in the P-site to release the nascent polypeptide from the stalled ribosome, whereas RF1 does not have this function. Thus, ArfA acts as an RF2 modulating factor that makes RF2 stop-codon-independent. Unlike *trans*-translation, ArfA does not add a degradation tag to the nascent polypeptide, so unnecessary polypeptides accumulate inside the cell. ArfA recognizes the stalled ribosome by a similar mechanism to SmpB: The C-terminal tail of ArfA binds to the vacant mRNA path downstream of the A-site (mRNA entry channel) of the stalled ribosome [34]. ArfA and RF2 bind to the ribosome independently rather than in complex with each other.

*E. coli* arfA mRNA has a specific cleavage site for RNase III within the coding region, which produces a non-stop mRNA [35,36]. Consequently, the cellular level of ArfA is kept very low due to *trans*-translation. When the *trans*-translation activity decreased, which can be caused by lack of a *trans*-translation component or too high demand for ribosome rescue to catch up in the cell, C-terminally truncated but functional ArfA becomes accumulated. Thus, ArfA is considered as the backup system of *trans*-translation at least in *E. coli*.

ArfA is narrowly distributed only within a subset of β- and γ-Proteobacteria. ArfT was found in *F. tularensis* as the second RF-dependent ribosome rescue factor [37]. It works with not only RF2 but also RF1. ArfT is more widely distributed among bacterial domain. Almost no sequence homology was found between ArfA and ArfT. Unlike *E. coli*, the *arfT* gene does not include an RNase III cleavage site and there is no difference of expression level of *arfT* mRNA between wild type and *ssrA* lacking cells. BrfA is found in *Bacillus subtilis* as an RF-dependent ribosome rescue factor that works with RF2 but not with RF1, whereas it has little sequence homology with ArfA or ArfT [38]. BrfA is limited to the Bacillus genus. Since the *brfA* gene contains a transcription termination sequence within the coding region instead of RNase III cleavage site, BrfA is synthesized from nonstop mRNA and its expression is regulated by *trans*-translation.

In the process of canonical termination, the decoding center nucleotides adopt a distinctive conformation that has not seen in elongation steps (Figure 2) [39,40,41,42]. G530 stacks on the third nucleotide of a stop codon. While A1492 flips toward G530, A1493 stacks on A1913 within h44. On the other hand, ArfA and RF2 binding causes a dramatic conformational change in the decoding center nucleotides [43,44,45,46,47]. G530 interacts with E30 of ArfA rather than stop codon. R26 of ArfA prevents A1492 from flipped-out conformation, resulting in stacking with A1913 within h44. Instead, A1493 adopts a flipped-out conformation. Essentially the same conformations of the decoding center are observed in pre-accommodation complex with ArfA A18T (loss of function mutant) and the compact form of RF2 [43]. Interestingly, in the BrfA-RF2-70S complex, the stacking interaction between A1493, instead of A1492, and A1913 is observed [38]. This is different from the conformation of the decoding center in termination complex but quite similar to that in TnaC·RF2 complex [48].

RF2 may act as a stop-codon-independent release factor even without special factors. In the elongation process, cognate aminoacyl-tRNA in complex with EF-Tu and GTP successfully enters the vacant A-site based on the codon-anticodon base pairing, while near-cognate aminoacyl-tRNA is strictly rejected through the dual selection system including the initial selection step and the proofreading step [49,50]. If an incorrect aminoacyl-tRNA enters the vacant A-site by escaping these steps and undergoes peptidyl transfer, an additional quality control system involving RF2 and RF3 operates: RF2 enters the vacant A-site in a codon-independent manner and hydrolyzes the peptidyl-tRNA bond to release the nascent polypeptide [51].

## 4. RF Homologue That Acts as a Stop-Codon-Independent RF

In 2011, another type of ribosome rescue factor, ArfB (YaeJ) was identified in *E. coli* [52,53]. The lethal phenotype of the double mutant simultaneously deleting tmRNA and ArfA can be suppressed by overexpression of ArfB. ArfB consists of the N-terminal globular domain and the C-terminal domain connected by a flexible linker region. The N-terminal globular domain contains the catalytic Gly-Gly-Gln (GGQ) motif, which is conserved in class I polypeptide release factor, but lacks a stop codon recognition motif. Therefore, ArfB enters the A-site of nonstop mRNA, in which mRNA-free entry channel is recognized by the C-terminal tail of ArfB, and it hydrolyzes the peptidyl-tRNA in the P-site as the stop-codon-independent RF. ArfB is widely distributed among bacteria except *Deinococcus-Thermus* or *Firmicutes* [52]. Mitochondria have ArfB homologues, ICT1 (Pth4) and C12orf65 (Pth3) [54,55].

In the ArfB-ribosome complex, the nucleobases of rRNA in the decoding center adopt distinct conformations. According to recent high-resolution 2.6 Å cryo-EM structure of ArfB-ribosome complex with mRNA+9 extending nine nucleotides from the P-site, A1493 is partially flipped out and stacked on A1913 and P110 of ArfB, while A1492 remains within h44 [56]. While A1492 adopts an *anti*-conformation, A1493 does a *syn*-conformation. These conformations of nucleobases in the decoding center are different from the crystal and other cryo-EM structures with 3.2 and 3.5 Å resolution, respectively, in which A1492 and A1493 adopt *syn*- and *anti*-conformations, respectively [57,58]. G530 contacts with the side chain of L119 via stacking interaction. Although the binding site of the C-terminal tail overlaps with that of the A-site codon on the ribosome, ArfB catalyzes hydrolysis of peptidyl-tRNA on the mRNA+9 complex [56,58].

Unlike *trans*-translation, this kind of ribosome rescue by RF homologue as well as RF-dependent ribosome rescue systems leaves incomplete, probably unnecessary, and in some cases harmful polypeptides inside the cell.

## 5. Ribosome Rescue after the Stalled Ribosome Is Split into Subunits

In eukaryotes, the stalled ribosome is split into subunits by Pelota/Dom34, Hbs1 and ABCE1, and the resulting large subunit retaining peptidyl-tRNA, but lacking mRNA, is rescued by the RQC (ribosome-associated quality control) system. Therein, tRNA^Ala^ and tRNA^Thr^, with the aid of RQC2, add alanine and threonine residues to the C-terminus of the nascent polypeptide, recruiting the E3 ligase Ltn1/listerin for ubiquitin/proteasome-dependent proteolysis [59]. A similar system was found in *B. subtilis* in which the large subunit with peptidyl-tRNA but without mRNA split from the stalled ribosome is rescued by tRNA^Ala^ together with RqcH, an RQC2 homologue and its partner protein RqcP (YaoB), producing a chimeric polypeptide comprising the mRNA-derived truncated polypeptide and the C-terminal poly alanine tail [60,61,62]. Since the C-terminal poly alanine tail serves as a degradation signal for Clp proteases, the products do not accumulate in the cell. The situation is quite similar to that of *trans*-translation in terms of involvement of aminoacylated RNA and production of a chimeric polypeptide for degradation.

Hsp15, which is known as a heat shock protein in *E. coli*, is a homologue of RqcP but having a longer C-terminal residues [61]. Although it binds to the large subunit with peptidyl-tRNA, Hsp15 is thought to have a distinct function from RqcP, as RqcH is absent in *E. coli*.

HflX, a heat shock-induced GTPase, has been found as a ribosome-splitting factor [63]. A gene encoding HflX is required for cell survival upon heat stress in *E. coli* [64]. It promotes splitting of the stalled ribosome with deacylated tRNA in the P-site. Thus, it is likely to act on the stalled ribosome after hydrolyzing the P-site peptidyl-tRNA by ArfA/RF2 or ArfB rather than that before RqcH/RqcP-dependent poly alanine tagging. Since Pelota/Dom34 homologue is absent in bacteria, how and when the bacterial stalled ribosome is split into subunits for RqcH/RqcP-dependent poly alanine tagging is yet to be investigated.

Besides ICT1, mitochondria have another kind of RF homologue having a GGQ motif but lacking a stop codon-recognition domain, C12orf65 (mtRF-R). It was recently found to be involved in mitochondrial ribosome-associated quality control acting with MTRES1 on the large subunit split from the stalled ribosome [65].

## 6. EF-P

EF-P was found as an elongation factor that promotes *N*-formylmethionyl-puromycin synthesis in vitro [66]. EF-P is universally conserved in bacteria and plays a role in a variety of cellular events [67,68]. In 2013, two groups have reported that EF-P is required for translation of consecutive proline codons [69,70]. Among 20 amino acids, proline is a poor substrate for peptidyl transfer reaction so that the ribosome often stalls at consecutive proline residues [71]. In this case, EF-P, which structurally mimics aminoacyl-tRNA, enhances peptide bond formation by binding to the region between the P- and E-sites, in which the hypermodified lysine residue of EF-P mimicking the aminoacyl moiety stimulates the catalytic center in the stalled ribosome. This allows the stalled ribosome to continue peptide elongation on the original mRNA.

## 7. EF4 (LepA)

EF4 (LepA) is highly conserved GTPase found in bacteria, mitochondria and chloroplasts. It has six domains I, II, III, IV, V and VI. Four (I, II, III and V) out of six domains are conserved in EF-G. EF4 possesses a specific C-terminal domain (VI), which is flexible and disordered in the crystal structure of ribosome-free EF4. Although the deletion of the gene from the *E. coli* genome shows no obvious phenotype in LB medium [72], the growth is affected under stressful conditions [73]. When the translating ribosome stalls after defective translocation under high concentration of intracellular magnesium ions or low temperature, EF4 binds to the post-translocation ribosome complex to facilitate back translocation of tRNAs from the E and P-sites to the P and A-sites, respectively, allowing the stalled ribosome to continue translation [74]. Ribosome profiling coupled with next-generation sequencing showed that lack of EF4 causes ribosome arrest at a glycine codon in the A-site [75]. However, a single-molecule Forster resonance energy transfer (smFRET) study shows that EF4/GTP prefers the pre-translocation state rather than the post-translocation state, arguing against the proposal of back-translocation [76].

Structural studies revealed that the C-terminal domain of EF4 reaches into the peptidyl transferase center [75,77]. A crystal structure of the 70S ribosome bound to EF4 with the nonhydrolyzable GTP analog GDPCP shows that it contains three tRNAs in the A-site, P-site and E-site resembling the pre-translocation state [75]. In this state, EF4 interacts with the acceptor stem of the A-site tRNA to induce a distortion of A-site tRNA (A/L tRNA), positioning the CCA-end out of the peptidyl-transferase center. On the other hand, the structure of the ribosome with GDP-bound EF4 contains two tRNAs in the P-site and E-site [77]. While the ribosome with GDPCP-bound EF4 is in a classical state of racheting, the ribosome with GDP-bound EF4 is ratcheted clockwise. The clockwise racheting movement of EF4 in the opposite direction mediated by EF-G causes G530 to shift away from the decoding center and A1492/A1493 to be disordered.

## 8. Arrest Sequences on mRNA

Specific amino acid sequences or specific nucleotide sequences affect the rate of translation, resulting in accumulation of stalled ribosome. One example of the arrest sequences is SecM peptide [78]. The interaction of the C-terminal arrest sequence (FxxxxWIxxxxGIRAGP) of SecM with the ribosome exit tunnel causes a translational arrest, allowing the translation of downstream secA gene. Other types of arrest sequences, such as TnaC and MifM, have been reported [79,80].

In 2009, a ribosome profiling method was developed to analyze comprehensive and quantitative measurements of ongoing translation by deep-sequencing of ribosome-protected fragments (RPFs) [81]. This approach revealed not only the positions of the ribosomes on mRNAs but also the rates of translation elongation, enabling genome-wide search of arrest sequences [82]. An earlier study on bacterial ribosome profiling reported that ribosomes tend to pause at internal Shine–Dalgarno-like sequences [83], although such tendency is not observed in other studies [84,85]. A recent study reported that the sample preparation significantly affects the quality of data, and an improved ribosome profiling revealed that ribosomes tend to pause at Asp, Gly and Pro codons [86], in agreement with previous biochemical studies [69,87]. The length of bacterial RPFs shows a broad distribution from 15 to 40 nt, which is in sharp contrast to the narrower distribution of yeast RPFs with a peak of 28 nt corresponding to the length of mRNA covered by a ribosome. It is tempting to think about why such a difference between eukaryotes and bacteria arises. Further studies are required to understand the arrest sites in vivo.

## 9. Recognition of Stalled Ribosome by Rescue Factors

It has been discussed how ribosome rescue factors recognize the target ribosome. SmpB, ArfA, BrfA, ArfB and ICT1 have an unstructured C-terminal tail that binds to the mRNA entry channel free of mRNA, so that they can find the ribosome stalled at the 3′ end of truncated mRNA. If the mRNA entry channel is occupied by mRNA, the C-terminal tail of these rescue factors would not interact with their binding sites on the ribosome. With regard to tmRNA·SmpB, while the rate of peptidyl-transfer by Ala-tmRNA·SmpB decreases as the mRNA 3′ extension increases [88,89], the rate of GTP hydrolysis by EF-Tu (preceding peptidyl transfer) is not affected by the length of mRNA [90]. In other words, while the tmRNA·SmpB·EF-Tu·GTP complex binds to the ribosome and activates GTP hydrolysis by EF-Tu regardless of mRNA length, it receives nascent polypeptide from the P-site peptidyl-tRNA only when the mRNA entry channel is empty. The situation is similar to the case of ArfA/RF2 pathway: ArfA and RF2 sequentially bind to the ribosome irrespective of the length of mRNA; however, they hydrolyze the peptidyl-tRNA only when ribosome is stalled on a truncated mRNA [34]. These results suggest that both tmRNA·SmpB and ArfA/RF2 recognize the target ribosome at the step after ribosome binding rather than initial binding step. Since the mRNA entry channel wraps around the neck region of the 30S subunit, the initial binding of ribosome rescue factors may induce a conformational change of the 30S subunit to open the mRNA entry channel for loading the C-terminal tail of rescue factors in the channel. Such a conformational change in the channel has been observed in the initiation and tmRNA complexes [91,92].

Association of ribosome-binding proteins with the mRNA entry channel may be a reasonable strategy to switch the mode of translation. In eukaryotes, Hbs1 binds to the mRNA entry channel to split the stalled ribosome into subunits with Pelota/Dom34 and ABCE1 [93]. Another example is translation inhibitor: Stm1 (SERBP1) and SARS-CoV-2 Nsp1 in eukaryotes inhibit translation by competing for mRNA binding to the mRNA entry channel [94,95,96,97]. Although there is no sequence homology between these mRNA channel-binding proteins, they are rich in basic amino acid residues. The pI values of the C-terminal regions of SmpB, ArfA and ArfB are over 10, whereas those of the N-terminal regions are under 10 (Table 1). High pI value in the C-terminal region is also observed in Stm1 but not in release factor and elongation factors that do not bind to the mRNA entry channel. Since the mRNA entry channel is primarily formed by 16S rRNA composed of negatively charged 16S rRNA, it is reasonable that the mRNA entry channel-binding proteins bind to the channel via positively charged residues of the proteins. Nsp1 is an exceptional case in which the C-terminal domain has a low pI value. Interactions other than electrostatic interactions might be more important between the C-terminal domain of Nsp1 and the mRNA entry channel.

## 10. mRNA Cleavage on the Ribosome

In bacteria, some ribosome-dependent RNases such as RelE, YoeB, YafQ and HigB that cleave the A-site codon have been reported [98,99,100,101]. These nucleases, belonging to a subclass of type II toxins of toxin–antitoxin systems, bind to the A-site on the ribosome and cleaves mRNA after the second position of the A-site codon. Although their activities are blocked by forming a complex with antitoxin, the toxin is activated by proteolytic digestion of antitoxin by stress-induced proteases such as Lon, resulting in accumulation of stalled ribosomes on cleaved mRNAs. The finding of A-site endonucleases is consistent with biochemical studies, in which tmRNA·SmpB, ArfA/RF2 and ArfB prefers ribosomes stalled at the 3′end of truncated mRNA rather than those stalled in the middle of intact mRNA [33,34,56,58,88,89,90,102]. A-site endonuclease has also been found in eukaryotes [103]. Cue2 nuclease is recruited to the collided ribosome on problematic mRNA to cleave mRNA within the A-site, allowing ribosome rescue by the ribosome quality control system. Interestingly, the Cue2 homolog SmrB was shown to target the collided ribosomes in *E. coli* [104]. Although both Cue2 and SmrB have an SMR (small MutS related) hydrolase domain, they have different target sites: Cue2 targets the A-site of the collided ribosome, while SmrB targets the 5′ boundary of the collided ribosome. SmrB homologues is likely to be widely distributed among the bacterial domain.

## 11. Conclusions

Various kinds of ribosome rescue systems operate even in a bacterial cell. Among them, only *trans*-translation always exists in the cell. The *E. coli* cell has two additional ribosome rescue systems, ArfA/RF2 and ArfB systems. Apparently, *trans*-translation has an advantage over ArfA or ArfB system in that it does not accumulate nonfunctional products in the cell. Thus, it is reasonable that most or probably all of the bacterial cells adopt *trans*-translation as the main ribosome rescue system. Bacteria have at least three different kinds of RF-dependent ribosome rescue factors that may be evolutionarily unrelated, while ArfB or its orthologue sporadically appears in the bacterial domain and also in mitochondria (Phylogenetic distribution of ribosome rescue factors is reviewed in Ref. [105]). It is tempting to think about why multiple ribosome rescue systems coexist in a cell. They may be used properly depending on the situation. For example, ArfA and ArfB systems may be more advantageous especially under energy-depleted conditions in that they do not require energy consumption for tagging and proteolysis. It is also interesting to think about how they have evolved.

In the past decade, X-ray crystal and cryo-EM structures revealed how ribosome rescue factors deal with the stalled ribosome at an atomic level (Figure 3). The structures provide information of the conformational rearrangements of the ribosome rescue factors and ribosomes. In particular, the nucleotides of 16S and 23S rRNA in the decoding center adopt distinct conformations that have not seen in the canonical translation to interact with ribosome rescue factor, SmpB, ArfA or ArfB. It has been suggested that the tRNA selection and the stop codon-dependent peptide release in canonical translation are highly dependent on an induced-fit mechanism. A similar mechanism would also underlie the ribosome rescue systems.

Recent extensive studies using next-generation sequencing technology revealed various arrest sequences and novel ribosome rescue factor candidates in various organisms [37,83,85,86]. However, further studies are needed to understand how the cell utilizes the different ribosome rescue factors in each situation.

## Figures and Tables

**Figure 1 microorganisms-10-00372-f001:**
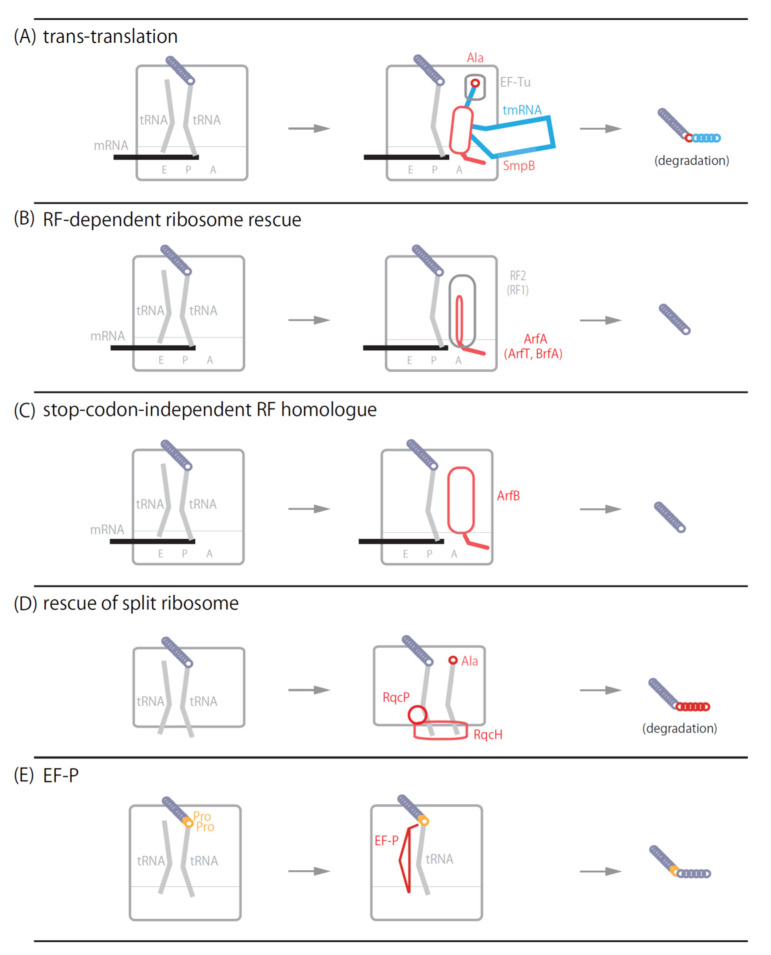
Schematic model of ribosome rescue systems. (**A**) A model of *trans*-translation. Ala-tmRNA·SmpB·EF-Tu·GTP quaternary complex enters the vacant A-site of the ribosome stalled on nonstop mRNA. Ala-tmRNA·SmpB receives the nascent polypeptide from the P-site peptidyl-tRNA and adds several amino acids to the C-terminus of the polypeptide as the degradation tag. (**B**) A model of RF-dependent ribosome rescue. ArfA, ArfT or BrfT with release factor RF2 (RF1 is also available in *Francisella tularensis* ArfT) binds to the ribosome stalled on nonstop mRNA to hydrolyze the ester bond between nascent polypeptide and P-site tRNA. (**C**) A model of stop codon-independent RF homologue. ArfB binds to the ribosome stalled on nonstop mRNA to hydrolyze the P-site peptidyl-tRNA. (**D**) A model of RqcH-RqcP system. RqcH and RqcP bind to the 50S subunit-peptidyl-tRNA complex. RqcH and RqcP adds poly-alanine tag to the C-terminus of nascent polypeptide as a degradation tag. (**E**) A model of EF-P mediated system. EF-P binds to the E-site to promote peptidyl transfer reaction on consecutive proline codons.

**Figure 2 microorganisms-10-00372-f002:**
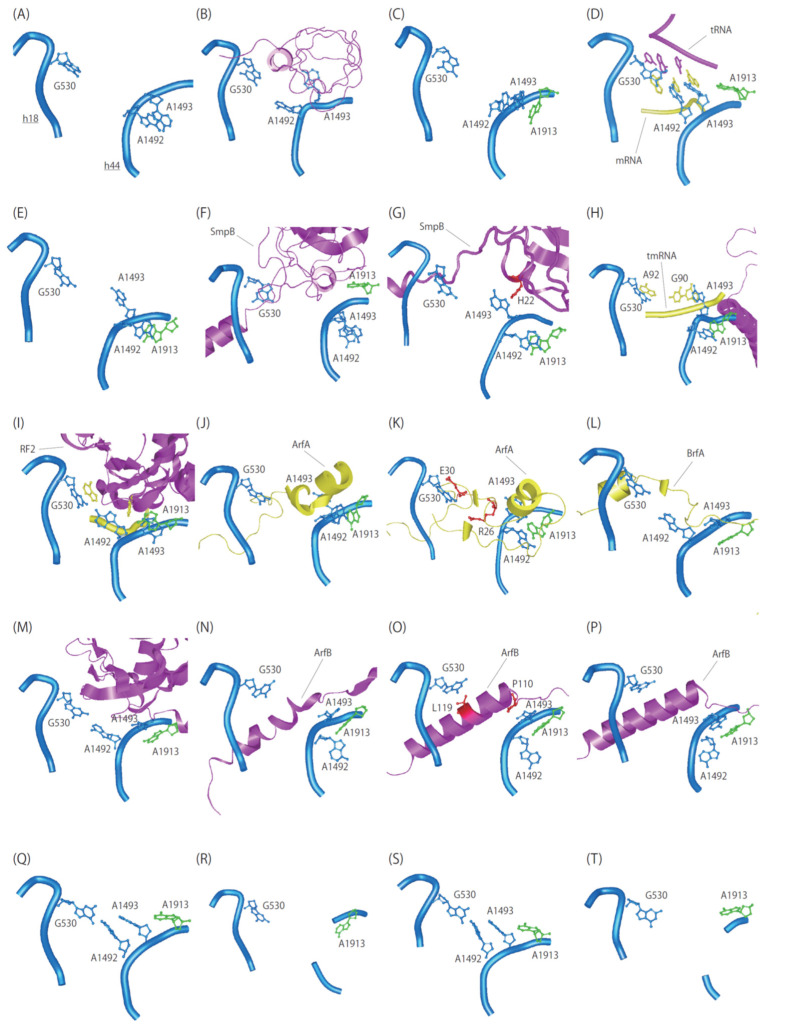
Conformation of rRNA and ribosome rescue factor in the decoding center. 16S and 23S rRNA nucleotides are shown in cyan and green, respectively. (**A**) Vacant A-site of the 30S subunit (PDBID:1J5E). (**B**) IF1 (magenta) bound to the 30S subunit (PDBID:1HR0). (**C**) Vacant A-site of the 70S ribosome (PDBID:4YBB). (**D**) Cognate tRNA (magenta) and mRNA codon (yellow) in the A-site of the 70S ribosome (PDBID:7K00). (**E**) Truncated mRNA bound to the 70S ribosome (PDBID:5MDZ). (**F**) Pre-accommodated complex of tmRNA·SmpB·EF-Tu·GDP bound to the 70S ribosome in the presence of kirromycin (PDBID:7ABZ). SmpB is shown in magenta. (**G**) Accommodated complex of tmRNA·SmpB bound to the A-site of the 70S ribosome (PDBID:7AC7). SmpB is shown in magenta. (**H**) Translocated complex of tmRNA·SmpB bound to the P-site of the 70S ribosome (PDBID:7ACJ). SmpB and resume codon of tmRNA are shown in magenta and yellow, respectively. (**I**) Release factor RF2 (magenta) bound to a stop codon (yellow) in the A-site of the 70S ribosome (PDBID:4V67). (**J**) Pre-accommodated complex of ArfA A18T (loss of function) and RF2 bound to the 70S ribosome (PDBID:5MDW). ArfA and RF2 are shown in yellow and magenta, respectively. (**K**) Accommodated complex of ArfA and RF2 bound to the A-site of the 70S ribosome (PDBID:5MDV). ArfA and RF2 are shown in yellow and magenta, respectively. (**L**) Accommodated complex of BrfA and RF2 bound to the 70S ribosome (PDBID:6SZS). BrfA and RF2 are shown in yellow and magenta, respectively. (**M**) RF2 (magenta) complex bound to the TnaC-mediated 70S ribosome (PDBID:7O1C). (**N**) *E. coli* ArfB (magenta) bound to the *Thermus thermophiles* 70S ribosome (PDBID:4V95). (**O**) *E. coli* ArfB (magenta) bound to the *E. coli* 70S ribosome (PDBID:6YSS). (**P**) *E. coli* ArfB (magenta) bound to the P/E hybrid tRNA with *E. coli* 70S ribosome (PDBID:6YST). (**Q**) LepA/GDPCP bound to the 70S ribosome (PDBID:5J8B). LepA is omitted for clarity. (**R**) LepA/GDP bound to the 70S ribosome (PDBID:4W2E). LepA is omitted for clarity. (**S**) Pre-translocated complex of EF-G (magenta) bound to the 70S ribosome (PDBID:4WPO). (**T**) Post-translocated complex of EF-G (magenta) bound to the 70S ribosome (PDBID:4WQF).

**Figure 3 microorganisms-10-00372-f003:**
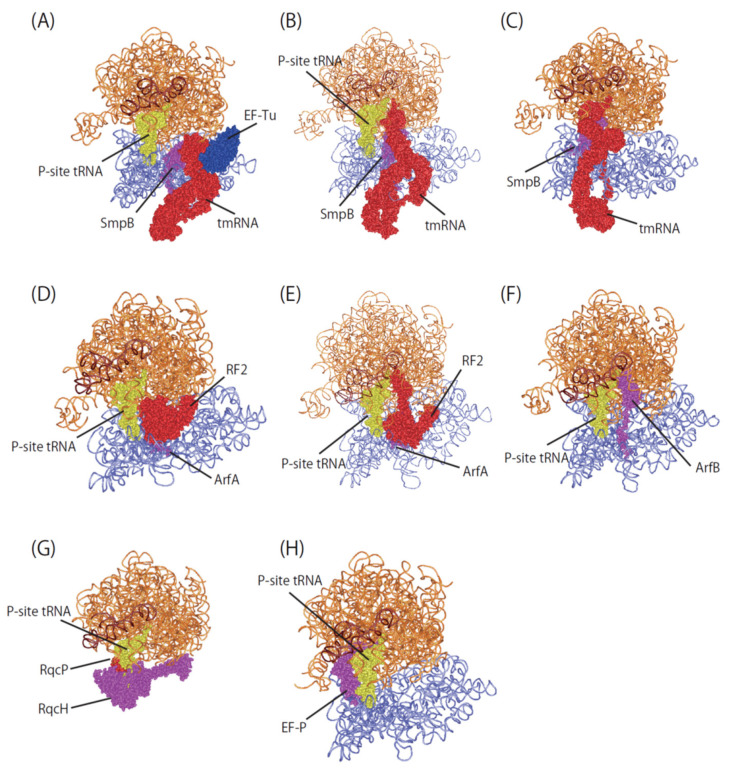
Overview of ribosome rescue factor complexes. (**A**) Pre-accommodation complex of Ala-tmRNA·SmpB·EF-Tu·GDP·kirromycin with 70S ribosome (PDBID:7ABZ). (**B**) Accommodation complex of tmRNA·SmpB with 70S ribosome (PDBID:7AC7). (**C**) Translocated complex of tmRNA·SmpB with 70S ribosome (PDBID:7ACJ). (**D**) Pre-accommodated complex of ArfA (A18T)/RF2 with 70S ribosome (PDBID:5MDW). (**E**) Accommodated complex of ArfA/RF2 with 70S ribosome (PDBID:5MDV). (**F**) ArfB complex with 70S ribosome (PDBID:6YSS). (**G**) RqcH/RqcP complex with 50S subunit (PDBID:7AS8). (**H**) EF-P complex with 70S ribosome (PDBID:6ENU).

**Table 1 microorganisms-10-00372-t001:** Isoelectric point (pI) of protein domains.

Protein	Domain	pI
*Escherichia coli* MG1655 SmpB	N-terminal domain (1–130)	9.58
	C-terminal tail (131–160)	10.37
*Escherichia coli* MG1655 ArfA	N-terminal region (1–25)	6.70
	C-terminal region (26–55)	11.13
*Escherichia coli* MG1655 ArfB	N-terminal domain (1–100)	8.16
	C-terminal tail (101–160)	12.11
*Escherichia coli* MG1655 RF2	Domain I (1–106)	4.02
	Domain II (107–208)	4.79
	Domain III (209–300)	6.38
	Domain IV (301–360)	5.24
*Escherichia coli* MG1655 EF-G	Domain I (7–279)	5.08
	Domain II (280–405)	4.58
	Domain III (408–484)	4.77
	Domain IV (485–609)	8.75
	Domain V (610–699)	4.94
*Saccharomyces cerevisiae* Stm1	Domain I (1–153)	9.07
	Domain II (154–218)	6.48
	Domain III (219–273)	12.23
SARS-CoV-2 Nsp1	N-terminal domain (1–127)	5.94
	C-terminal domain (128–180)	4.63

## Data Availability

All data are available in the manuscript.

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
