# Peer review of "Bacterial Ribosome Rescue Systems"

_microorganisms, 2022, doi:10.3390/microorganisms10020372_

Round 1
Reviewer 1 Report
This review summarizes the various ribosome rescue systems identified in bacteria, including tmRNA, Arf, RQC, etc. The review discusses the mechanism, distribution, and structural insights of these ribosome rescue systems. It should be of interest to a broad readership of Microorganisms.
Minor comments:
- It would be helpful to include a table describing the phylogenetic distribution of different rescue systems discussed in the manuscript.
- Please add references: Line 18, “…stalled ribosomes”
- Typo: Line 23, “1990th” should be “1990s”;
- Italicize gene names, e.g., Line 137 arfT
- Line 169, reword the sentence starting with “The lethal phenotype…”
Author Response
- It would be helpful to include a table describing the phylogenetic distribution of different rescue systems discussed in the manuscript.
Response 1: A nice figure describing the phylogenetic distribution of different rescue systems appears in “Burroughs & Aravind, Int. J. Mol. Sci. 20 (2019)”. We cited it (ref. 105) instead of making a new Table.
- Please add references: Line 18, “…stalled ribosomes”
Response 2: Some references are added.
- Typo: Line 23, “1990th” should be “1990s”;
Response 3: Thank you for suggesting our mistake. It is corrected.
- Italicize gene names, e.g., Line 137 arfT
Response 4: Gene names are italicized.
- Line 169, reword the sentence starting with “The lethal phenotype…”
Response 5: Thank you for suggesting our mistake. It is corrected.
Reviewer 2 Report
The manuscript "Bacterial Ribosome Rescue Systems" by Daisuke Kurita and Hyouta Himeno is a conscise review, highly focused and easy to follow. All relevant information on a complete range of ribosome rescue systems in bacteria is included with necessary and appropriate reference to the eukaryotic rescue systems. I consider this manuscript to be nearly ideal short review on this topic. My only suggestion is to present an additional figure illustrating not only the decoding center of the ribosome intaracting with rescue factors, but also a kind of structural overview of an entire ribosome*rescue factor complexes. These structures are so beatiful so they deserve to be shown.
Author Response
Thank you for your invaluable comments. We added a new figure illustrating structural view of the ribosome in complex with each rescue factor.